# Transdermal Fentanyl in Patients with Cachexia—A Scoping Review

**DOI:** 10.3390/cancers16173094

**Published:** 2024-09-05

**Authors:** Andrea Carlini, Emanuela Scarpi, Carla Bettini, Andrea Ardizzoni, Costanza Maria Donati, Laura Fabbri, Francesca Ghetti, Francesca Martini, Marianna Ricci, Elisabetta Sansoni, Maria Valentina Tenti, Alessio Giuseppe Morganti, Eduardo Bruera, Marco Cesare Maltoni, Romina Rossi

**Affiliations:** 1Medical Oncology, IRCCS Azienda Ospedaliero-Universitaria di Bologna, 40126 Bologna, Italy; andrea.carlini10@studio.unibo.it (A.C.); andrea.ardizzoni2@unibo.it (A.A.); 2Department of Medical and Surgical Sciences, University of Bologna, 40126 Bologna, Italy; costanzamaria.donati@unibo.it (C.M.D.); alessio.morganti2@unibo.it (A.G.M.); marcocesare.maltoni@unibo.it (M.C.M.); romina.rossi10@unibo.it (R.R.); 3Biostatistics and Clinical Trials Unit, IRCCS Istituto Romagnolo per lo Studio dei Tumori (IRST) “Dino Amadori”, 47014 Meldola, Italy; 4Palliative Care Unit, Azienda Unità Sanitaria Locale (AUSL) Romagna, 47121 Forlì, Italy; carla.bettini@studio.unibo.it (C.B.); laura.fabbri@auslromagna.it (L.F.); francesca.ghetti@auslromagna.it (F.G.); francesca.martini@auslromagna.it (F.M.); elisabetta.sansoni@auslromagna.it (E.S.); mariavalentina.tenti@auslromagna.it (M.V.T.); 5Radiation Oncology, IRCCS Azienda Ospedaliero-Universitaria di Bologna, 40126 Bologna, Italy; 6Palliative Care Unit, IRCCS Istituto Romagnolo per lo Studio dei Tumori (IRST) “Dino Amadori”, 47014 Meldola, Italy; marianna.ricci@irst.emr.it; 7Department of Palliative Care, Rehabilitation and Integrative Medicine, The University of Texas MD Anderson Cancer Center, Houston, TX 77030, USA

**Keywords:** fentanyl, transdermal patch, cachexia, body mass index, albumin, pharmacokinetics

## Abstract

**Simple Summary:**

This scoping review explores the use of transdermal fentanyl (TDF) for pain management in patients with cachexia, which is a severe wasting syndrome associated with cancer and other advanced illnesses. While TDF is commonly used in the management of chronic cancer-related and non-cancer related pain, its efficacy and safety in cachectic patients remain unclear due to altered pharmacokinetics (PK) in these individuals. This review examines nine studies that analyzed the impact of cachexia on the efficacy and tolerability of TDF. The findings suggest mixed results: some studies showed that cachexia could reduce TDF effectiveness and increase the required dose, while others found little to no impact or even potential improvement in outcomes. The current evidence is insufficient to provide definitive guidelines for the use of TDF in cachectic patients, highlighting the need for further research in this area.

**Abstract:**

Cachectic patients frequently require transdermal fentanyl (TDF) for pain management, but data on its efficacy and safety are scarce and inconsistent. This scoping review aims to analyze the evidence concerning TDF administration in patients with cachexia irrespective of the underlying pathology. The primary objective is to assess the analgesic efficacy and tolerability of TDF in cachectic patients. The secondary objective is to identify cachexia characteristics that may influence fentanyl pharmacokinetics (PK). A comprehensive search of PubMed, Embase, and Web of Science databases was conducted up to March 2024. The review included observational and clinical studies on cachectic patients with moderate to severe pain treated with TDF patches at any dosage or frequency. Phase 1 trials, animal studies, case reports, preclinical studies and conference abstracts were excluded. Nine studies were included: four studies reported that cachexia negatively impacted TDF efficacy, increasing required doses and lowering plasma concentrations; three studies found minimal or no impact of cachexia on TDF efficacy and PK; two studies suggested that cachexia might improve TDF outcomes. Study quality ranged from moderate to high, according to the National Institutes of Health (NIH) Quality Assessment Tool. The current evidence is insufficient to provide any definitive recommendations for TDF prescribing in cachectic patients.

## 1. Introduction

Fentanyl is a synthetic µ-opioid receptor agonist with an affinity 80–100 times greater than morphine, which is available by various routes of administration [1]. Fentanyl transdermal formulations are used for the sustained relief of chronic cancer-related [2] and non-cancer-related pain [3]. Some studies have suggested that transdermal fentanyl (TDF) is associated with a lower incidence of adverse effects such as constipation, nausea, vomiting and daytime drowsiness [4]. A systematic review comparing TDF and buprenorphine with other oral opioids for moderate to severe cancer pain found no significant differences in efficacy, but they noted better results for constipation and patient preference with transdermal formulations [5]. Consequently, TDF is one of the most frequently prescribed opioids globally, and it is recommended for patients with moderate to severe cancer pain on stable opioid therapy [6].

Fentanyl’s high lipophilicity and volume of distribution (300–400 L/70 kg) facilitate rapid transdermal absorption, creating a cutaneous depot in the stratum corneum. Subsequently, the drug passively diffuses to the dermis, where it is removed by cutaneous microcirculation [7]. The mean bioavailability of TDF is 92% (57–146%) [8]. Serum fentanyl concentrations increase over 12–14 h after the first patch application and reach steady state during the first day of the second patch administration [9]. The maximum serum concentration (C_max_) ranges from 0.7 μg/L (25 μg/h patch) to 2.6 μg/L (100 μg/h patch), whereas the time between patch application and the occurrence of C_max_ (t_max_) ranges from 12 to 48 h [7]. Fentanyl binds predominantly to albumin (Alb) in plasma (95%) [10] and is mainly metabolized by the hepatic cytochrome P450 enzymes CYP3A4 and CYP3A5 into inactive metabolites with less than 10% excreted unchanged in urine. The area under the curve (AUC; 33–126 μg/L h) is significantly increased with hepatic impairment, whereas renal impairment has a minimal effect [7,11]. After discontinuation, fentanyl has a slow elimination half-life averaging about 17 h [7].

Cachexia is a multifactorial, often irreversible wasting syndrome associated with cancer and other chronic illnesses such as AIDS, heart failure, kidney disease, COPD, cystic fibrosis, rheumatoid arthritis, Alzheimer’s disease, and infectious diseases. Over 80% of advanced cancer patients display cachexia, which contributes significantly to cancer mortality [12]. Cachexia’s pathophysiology includes systemic inflammation, muscle and adipose tissue depletion, decreased appetite, and altered metabolism [13]. The European Society for Medical Oncology (ESMO) defines cancer cachexia as a disease-related malnutrition subtype with specific phenotypic and systemic inflammation criteria [14].

The well-documented inter- and intra-individual variability in fentanyl pharmacokinetics (PK) [15] and analgesic effects [16] can be influenced by age, gender, liver and renal function, body temperature, genetic polymorphisms, and the use of CYP3A4 inducers/inhibitors [17,18,19,20]. In addition, Heiskanen et al. first suggested that cachexia and its pathophysiological changes may reduce serum fentanyl concentrations [21]. Since then, there remains a scarcity and heterogeneity of data on this topic. Recently, Davis addressed this issue in a letter to the editor, concluding that due to the less predictable PK in cachectic patients, it may be prudent to avoid TDF in individuals with advanced cachexia [22]. Therefore, there is an urgent need to better understand how cachexia interacts with fentanyl PK and alters its efficacy–safety profile.

This scoping review aims to analyze and categorize the evidence concerning TDF administration in patients with cachexia irrespective of the underlying pathology. The primary objective is to assess whether the efficacy and tolerability of TDF in cachectic patients differ from those observed in non-cachectic patients. The secondary objective is to determine whether the specific clinical effects of TDF in cachectic patients can be attributed to pharmacokinetic differences.

## 2. Materials and Methods

### 2.1. Search Strategy

This scoping review was conducted in accordance with the Preferred Reporting Items for Systematic Reviews and Meta-Analyses Extension for Scoping Reviews (PRISMA-ScR) guidelines [23]. A comprehensive literature search was conducted in PubMed, Embase and Web of Science databases up to March 2024. There were no restrictions on publication language. The search terms included fentanyl, transdermal patch, administration, cachexia, low body mass index, cancer-related weight loss, nutritional status, (hypo)albuminemia, and serum albumin. Detailed search terms are given in the Appendix A. Further eligible studies were identified by examining the reference lists of all eligible studies and published reviews. There has been no registration of the protocol, but it is available on request from the corresponding author.

### 2.2. Inclusion and Exclusion Strategy

We included observational and clinical studies that enrolled cachectic patients experiencing moderate to severe pain and using the TDF patch at any dose or frequency of application. We excluded phase 1 clinical trials involving healthy individuals, animal or laboratory models, ecological studies, cross-sectional studies, case reports, case series, editorials, letters to the editor, preclinical studies, and conference abstracts. To identify potentially eligible studies, two reviewers independently screened titles, abstracts and full-text articles. Disagreements were resolved by a third reviewer. Using standardized forms, two authors independently extracted the main study characteristics and outcomes of each eligible study. The main study characteristics included the following:Methods: study type, setting, timing of the evaluation.Participants: general definition of the enrolled population, site of cancer/other diseases, type, location and intensity of pain, number of subjects, mean/median age, sex/gender.Interventions: administered drug with dosage, titration, rescue drug, any dose increase/decreases, behavior in relation to any other drugs.Outcomes: We categorized the outcomes into primary and secondary. Primary outcomes describe the response to TDF in terms of efficacy and tolerability (NRS, Visual Analogue Scale [VAS], Edmonton Symptom Assessment Scale [ESAS], Symptom Distress Score [SDS], successful/partial successful opioid rotation, worst and least pain intensity scores, percent pain relief, number of rescue events, frequency of opioid-induced side effects), while secondary outcomes define the impact of fentanyl on some key PK stages of the drug (TDF dose, Morphine Equivalent Dose [MED], plasma fentanyl and norfentanyl concentration, Metabolic Ratio [MR], Transepidermal Water Loss [TEWL]).

### 2.3. Data Extraction and Synthesis

Data extraction involved collecting detailed information on study design, participant characteristics, interventions, and outcomes. The extracted data were synthesized to categorize the evidence concerning the PK, efficacy, and tolerability of TDF in cachectic patients. This synthesis aimed to identify patterns and gaps in the existing literature to inform future research and clinical practice.

### 2.4. Quality Appraisal of Evidence

The National Institutes of Health (NIH) Quality Assessment Tool for observational cohort and cross-sectional studies [24] was used to assess the quality of the included trials. The assessment focused on potential sources of bias, confounding variables, study power, and the robustness of causal relationships between interventions and outcomes. The NIH Quality Assessment Tool checklist was used to determine the overall risk of bias for each study, which was categorized into three levels: good (10–14), fair (6–9), or poor (0–5). Two independent reviewers performed the analysis. Discrepancies in assessment were resolved by discussion and consensus or by consultation with the senior author.

## 3. Results

### 3.1. Search Process

The selection process is detailed in the PRISMA-ScR 2020 flow diagram (Figure 1). Searches conducted on PubMed, Embase, and Web of Science databases, along with reference list examinations, identified a total of thirty-three papers. Nine articles were excluded as duplicates, and six additional articles were excluded after title–abstract screening due to irrelevance to the review question. Five articles could not be obtained in full-text format. Of the thirteen remaining articles, four were case reports and were excluded based on study design. Consequently, nine articles were included in the review [19,21,25,26,27,28,29,30,31].

The study by Clemens et al. included patients with chronic cancer pain who had been pre-treated with TDF and admitted to a Palliative Care (PC) unit, among whom 27 (33.3%) were cachectic. This study evaluated the impact of opioid rotation on pain and other symptoms, including cachexia. Although it provided valuable insights into TDF PK in “frail” and advanced patients, this article was excluded because it did not examine the role of TDF exposure specifically in cachectic patients or according to cachexia features [32].

### 3.2. Characteristics of the Studies

The characteristics of the studies are summarized in Table 1. All the studies were observational, consisting of three retrospective and six prospective studies. All included adult patients with different types of cancer. Only one study (Chiba et al.) involved non-oncological patients [31]. There was no homogeneity in malnutrition and/or cachexia definition: three studies relied solely on BMI [21,29,30], three used both BMI and serum Alb [19,25,28], one utilized the Nutritional Risk Screening 2002 (NRS-2002) questionnaire [26], and two applied the European Palliative Care Research Collaborative (EPCRC) criteria [27,31]. In five studies, the opioid conversion from or to TDF was investigated, as follows:From other opioids to 72 h TDF and the patch was maintained for 3 days at least [21].From Continuous Intravenous Infusion (CII) to 72 h TDF, using a 2-step taper over 6 h [25].From oxycodone to 72 h TDF [26].From other opioids to 72 h TDF [28].From oral oxycodone/morphine to 72 h TDF. Then, some patients were switched to morphine injection [31].

In the remaining four studies, a TDF patch was applied with no mention of previous opioid administration:72 h TDF, maintained for 3 days at least [19].24 h TDF patch, maintained for 3 days at least [27].72 h TDF patch, maintained for 8 days at least [29].72 h TDF patch [30].

All the studies allowed the use of rescue medications. Heiskanen et al. reported the use of laxatives for constipation and haloperidol for nausea [21], whereas Nomura et al. noted the occasional use of stool softeners, laxatives, or enemas [25]. Takahashi et al. and Chiba et al. permitted the use of Non-Steroidal Anti-Inflammatory Drugs (NSAIDs) [26,31]. Barratt et al. permitted and analyzed the use of CYP3A inhibitors–inducers [19], whereas Nomura et al. and Suno et al. excluded all patients receiving drugs that might affect the metabolism of CYP3A4 [25,27].

All the studies considered one or more primary outcomes (as previously defined in our review) except for Barratt et al. and Kuip et al., who focused exclusively on secondary outcomes (plasma fentanyl and norfentanyl concentration, MR) [19,29]. Among the primary outcomes, NRS was used in three articles [25,26,31]; frequency of opioid-induced side effects in three articles [25,27,28]; VAS in two articles [21,27]; number of rescue events [25], worst–least pain intensity scores, percent pain relief [30], ESAS, SDS and successful/partial successful opioid rotation [28] in one article. Among the secondary outcomes, plasma fentanyl concentration was used in five articles [19,21,25,27,29]; TDF dose was used in three articles [21,29,31]; MED was used in two articles [28,29]; plasma norfentanyl concentration, MR [25], and TEWL were used in one article [31].

### 3.3. Analysis of the Evidence

The impact of TDF on primary and secondary outcomes is summarized in Table 2. Four studies described a detrimental effect of cachexia on the efficacy–tolerability of TDF and/or its PK. One or more characteristics of cachexia negatively affected pain intensity scores, increased the mean TDF dose required for analgesia, and/or decreased plasma fentanyl concentration [21,25,26,31]. Heiskanen et al. found that patients with low BMI (BMI < 18 kg/m², *n* = 10) required significantly higher mean daily TDF doses than those with normal BMI (BMI 20–25 kg/m², *n* = 10) and exhibited lower plasma fentanyl concentrations at 48 and 72 h from baseline. Nevertheless, the pain intensity reported by patients was similar in both groups at baseline and at the end of the study. Additionally, no differences were observed in key modifiers of fentanyl absorption (local skin blood flow, skin temperature, and local sweating) except for a thinner upper arm skin fold in low BMI patients [21]. Nomura et al. included 18 patients that were converted from CII of fentanyl to TDF and measured the dose-adjusted serum fentanyl concentrations at at 3, 6, 9, 12, 15, 18, and 24 h after the baseline. Compared to baseline, there were no significant differences in pain intensity, number of rescue events, or opioid-related adverse events at any time point. Fentanyl concentrations decreased gradually after switching. However, there were no dose-related differences between the BMI groups at any time point. However, when Alb levels were considered, fentanyl concentrations at 9, 12, 15, 18, and 24 h were lower in the low Alb group (Alb < 3.5 g/dL, *n* = 9) than in the normal Alb group (Alb ≥ 3.5 g/dL, *n* = 9) [25]. Takahashi et al. found that the mean pain intensity in the low-nutrition group (NRS-2002 ≥ 3, *n* = 56) was significantly higher than that in the normal-nutrition group (NRS-2002 < 3, *n* = 36). An increase in NRS-2002 score was correlated with higher pain intensity with an odds ratio of 30.0 (95% CI, 4.48–200.97; *p*-value = 0.0005) [26]. Chiba et al. demonstrated that after switching from oral oxycodone/morphine, both fentanyl patch dose and MED were significantly higher in the cancer-cachexia group (*n* = 30) compared to the non-cancer-cachexia group (*n* = 47). In addition, the mean change in pain intensity was greater in the cancer-cachexia group. The average TEWL, an index of cutaneous dryness, was significantly lower in the cancer-cachexia group than in the non-cancer-cachexia group. For 9 out of 30 patients assigned to the cancer-cachexia arm, switching from TDF to morphine injection resulted in a reduction in both mean pain intensity and MED [31]. This finding is consistent with the study by Clemens et al., which demonstrated that switching from TDF to sustained-release (SR) morphine/hydromorphone significantly reduced pain scores and the frequency of rescue doses in patients admitted to a PC unit. Furthermore, adequate pain relief was achieved with lower equianalgesic morphine doses in the group of patients who switched from TDF to morphine [32]. In all these studies, the efficacy and safety of TDF, as well as the PK parameters, were adversely affected by the presence of one or more features of cachexia. The authors suggested that changes in skin permeability (xerosis [21], cutaneous dryness [26,31]) and low serum Alb levels [25] are the primary factors contributing to the poor absorption and performance of TDF in cachectic patients. Only Heiskanen et al. gave the clinical indication of avoiding TDF prescription in cachectic patients with pain [21].

Three studies reported minimal or no impact of cachexia on the efficacy and tolerability of TDF and/or its PK [19,28,29]. Barratt et al. conducted a comprehensive investigation involving a large cohort (*n* = 620) from the European Pharmacogenetic Opioid Study (EPOS), which is a multicenter collaboration aimed at identifying patient factors influencing opioid requirements for moderate to severe cancer pain. Significant inter-individual variability in fentanyl metabolism to norfentanyl was observed among patients receiving TDF patches. The primary determinants of fentanyl and norfentanyl concentrations were the fentanyl delivery rate for both, and, additionally, serum fentanyl levels for norfentanyl. Although low BMI and hypoalbuminemia were associated with higher MRs (= serum norfentanyl concentration/serum fentanyl concentration), increased norfentanyl concentrations and decreased fentanyl concentrations, their overall contributions were minimal (1.1% and 0.4% for norfentanyl concentration; 0.5% and 0.4% for fentanyl concentration, respectively). Thus, the effect on fentanyl PK was small, and the impact on clinical outcomes was not assessed [19]. Reddy et al. studied a cohort of patients who switched from other opioids to TDF (*n* = 129) in order to evaluate the success of the conversion and to determine the opioid rotation rate of oral MED to TDF. Five weeks after the baseline, 59% of patients underwent a successful opioid rotation. There were no significant independent predictors of successful opioid switching among the variables tested, including BMI and Alb levels. Additionally, the opioid rotation ratio, which measures the relative equianalgesic potency of opioids, did not significantly change in patients with lower Alb and BMI [28]. Kuip et al. performed a clinical study in patients using a stable dose of fentanyl, obtaining a blood sample for PK analysis one day after patch application. They found that dose-adjusted (dose of 25 μg/h) plasma concentrations in both the low BMI group (BMI < 20 kg/m², *n* = 20) and the high BMI group (BMI > 25 kg/m², *n* = 27) did not significantly differ from the normal BMI group (BMI 20–25 kg/m², *n* = 41). Alb levels were within normal ranges and comparable in all BMI groups [29]. These three studies did not provide any clinical indication about TDF prescription in cachectic patients.

Two studies described a paradoxical beneficial effect of cachexia on TDF PK and/or its efficacy–tolerability. One or more cachexia feature positively affected pain intensity scores or increased plasma fentanyl concentration [27,30]. Suno et al. demonstrated that following the application of a 24-h TDF patch, the dose-adjusted fentanyl concentration (dose of 25 μg/h) was significantly higher in patients with refractory cachexia (*n* = 4) compared to those with pre-cachexia (*n* = 8) without any difference in the VAS score for pain intensity between the groups. Multiple regression analysis identified three factors—aspartate transaminase (AST), CRP, and estimated glomerular filtration rate (eGFR)—that might influence the dose-adjusted concentration of fentanyl. Alanine transaminase (ALT), serum creatinine and Alb were not significantly affected by dose-adjusted plasma concentrations. Although fentanyl is recommended for moderate to severe pain in patients with chronic kidney disease [33], this study postulated that a reduction in the eGFR may increase fentanyl plasma concentrations. Additionally, the authors proposed that cachexia-associated systemic inflammation, as indicated by increased CRP levels, may decrease CYP3A4 levels by downregulating its expression, possibly leading to increased plasma fentanyl concentrations [27]. Moryl et al. divided a cohort of patients treated with TDF patches by using two BMI classifications: one with five BMI categories (<20, 20–21.9, 22–24.9, 25–27.9, ≥28) and one with four categories (underweight or BMI < 18.5, normal weight or BMI 18.5–24.9, overweight or BMI 25.0–29.9 and obese or BMI ≥ 30). When using the first classification, patients with BMI < 20 reported the most pain relief and least pain while receiving the lowest average TDF dose. When using the second classification, patients with cachexia reported the most pain relief and least pain while also receiving the lowest TDF dose. Furthermore, BMI category <18.5 (second classification) was associated with greater pain relief regardless of TDF dose. However, the authors did not suggest avoiding or reducing the prescription of fentanyl in cachectic patients, as no significant association was found between BMI and TDF dose [30].

### 3.4. Quality Appraisal of Evidence 

We assessed methodological quality applying a validated scoring system. Using criteria from the NIH Quality Assessment Tool for Observational Cohort and Cross-Sectional Studies, two authors independently assessed risk of bias for each study with disagreements resolved by discussion. Table 3 shows the risk of bias analyses. Overall, the studies included in this scoping review have a low risk of bias and were considered of adequate quality for inclusion in the review.

## 4. Discussion

Currently, there are no definitive guidelines regarding the prescription of TDF in cachectic patients. The European Association for Palliative Care (EAPC) and ESMO guidelines support the use of transdermal opioids as a viable alternative to oral opioids for the management of moderate to severe pain requiring stable opioid administration. This recommendation is particularly relevant for patients experiencing conditions common in cachexia, such as nausea, vomiting, swallowing difficulties, constipation, and poor compliance [6,34]. Nevertheless, TDF is not free from common opioid-induced adverse effects, such as sedation, coma, respiratory depression, numbness, dizziness, mental clouding, nausea, vomiting, delirium, hallucinations and addiction [35]. In the US, adverse drug events related to opioid analgesics account for 8.4% of Emergency Department (ED) visits [36]. Documented cases of opioid-induced adverse events in cachectic patients receiving TDF highlight the potential for ED admissions. For instance, a 73-year-old metastatic cancer patient treated with 100 μg/h TDF for pain management experienced significant weight loss and subsequently developed auditory and visual hallucinations, which resolved after discontinuation and adjustment of the TDF dose [37]. In another case, a 55-year-old woman with anorexia nervosa and other comorbidities was admitted for pain management of a severe sacral pressure ulcer. She applied an excessive number of fentanyl patches, resulting in respiratory depression and coma, which was reversed with naloxone administration. Despite a significant reduction in opioid dose post-intervention, she exhibited no withdrawal symptoms, suggesting suboptimal TDF absorption due to some alteration in fentanyl PK, which was probably related to cachexia [38]. These cases highlight the urgency to better comprehend how cachexia interacts with the TDF exposure–response relationship and to ensure effective pain management while minimizing severe adverse effects, particularly in the context of frail and often end-of-life patients.

To the best of our knowledge, our review is the first attempt to analyze and synthetize all the available evidence concerning the impact of cachexia on TDF efficacy, tolerability and PK irrespective of the underlying pathology. Regarding the primary objective of the review, the evidence gathered is insufficient to draw definitive conclusions about the efficacy and tolerability of TDF in cachectic patients. Four studies reported that cachexia negatively affected pain intensity scores, increased the mean TDF dose required for analgesia, and/or decreased plasma fentanyl concentration; three studies found minimal or no effect of cachexia on clinical and pharmacokinetic parameters; and two studies suggested that cachexia positively affected pain intensity scores or increased plasma fentanyl concentration. There is no consensus within these groups on the clinical impact of cachexia on pain control and opioid-related side effects. In the first group of studies, only Takahashi et al. and Chiba et al. found a significant effect on NRS [26,31]. The studies by Heiskanen et al. and Nomura et al., which showed no effect on pain, involved a small number of participants (*n* = 20 and 18, respectively), potentially limiting sample representativeness [21,25]. In the last group of studies, only Moryl et al. reported better pain control in patients with lower BMI [30]. Opioid-related adverse effects were similar in populations with and without cachexia [25,28]. Suno et al. used the rate of opioid-related adverse effects as a primary outcome but did not compare by cachexia status [27]. Given these conflicting findings, it is currently impossible to provide a definitive recommendation on whether, when, or how to prescribe TDF in these patients.

Regarding the secondary objective of the review, the authors only hypothesized possible explanations for the interaction between cachexia and TDF administration. In particular, the authors who suggested a negative role of cachexia focused on the absorption phase of TDF PK, which involves fentanyl diffusion through the keratinous stratum corneum and dermis, where the drug is removed by cutaneous microcirculation [7,8]. Heiskanen et al. observed comparable local skin blood flow, skin temperature, and local sweating levels between individuals with normal and low BMI except for a reduced thickness of upper arm skin folds in the latter group. Given the absence of disparities in skin microcirculation and the negligible impact of subcutaneous fat tissue on transdermal drug absorption, they postulated that alterations in skin permeability (xerosis) in cachectic patients could modify the stratum corneum’s diffusive capacity [21]. Similar conclusions were reached by Takahashi et al., who proposed that cutaneous dryness, a common clinical sign in cachectic patients [39], could impair TDF absorption [26]. A pre-clinical study observed that serum fentanyl concentrations in rats with dry skin were significantly lower than those in rats with normal skin [40]. Chiba et al. found that TEWL, reduced in case of dry skin, was significantly lower in cancer-cachexia patients than in non-cancer-cachexia patients [31]. Increased inflammatory cytokines in cancer cachexia, such as interleukin (IL)-1β, IL-6, and Tumor Necrosis Factor-α (TNF-α), cause lipid degradation and reduced lipid synthesis [13]. In a study on atopical dermatitis, IL-4, produced by T helper 2 (T_h2_) lymphocytes, decreases ceramide, which is a key lipid component in the skin’s water-holding properties [41]. This suggest a possible mechanism through which the systemic inflammatory status associated with cachexia could reduce fentanyl transcutaneous permeability and circulating levels. Nomura et al. highlighted the role of hypoalbuminemia, which is a key characteristic of cachexia syndrome [42]. Even without apparent edema or ascites, hypoalbuminemia could lead to undetectable edema and decreased skin permeability, affecting TDF absorption [25]. Conversely, authors proposing a beneficial role for cachexia emphasized the hepatic metabolism and renal excretion of TDF. Suno et al. found that reduced eGFR increased fentanyl plasma concentrations, contradicting recommendations for safe TDF use in chronic kidney insufficiency due to low unchanged fentanyl in urine and inactive metabolites [43]. This study also identified AST and CRP levels as significant contributors to increased serum fentanyl concentrations. Inflammatory cytokines in cachexia induce CRP production and inhibit Alb synthesis, which are positive and negative hepatic acute-phase proteins, respectively [13]. IL-6 reduces CYP3A4 messenger RNA (mRNA) levels and downregulates CYP3A4 expression, leading to decreased fentanyl metabolism and increased concentrations [44]. A study by Naito et al. found similar results for oxycodone, which is metabolized by CYP3A4 like fentanyl, suggesting increased plasma oxycodone concentrations in cachectic patients due to reduced CYP3A4 activity [45]. Unfortunately, the studies included in our review did not examine the relationship between immune status, cachexia, and opioid use. Notably, an emerging body of evidence suggests that opioids may interact with immune system activity in cancer patients [46], particularly in those undergoing treatment with immune checkpoint inhibitors (ICIs) [47]. These aspects, which could be crucial for a comprehensive risk–benefit analysis of opioid therapies, should be considered in future research. Given the limited evidence, the potential impact of cachexia on TDF PK appears to be multifaceted, involving altered skin permeability and systemic inflammatory responses that could modulate both drug absorption and metabolism, leading to significant variability in circulating fentanyl levels.

According to quality evaluation tools such as the NIH quality assessment tool, all included studies are of medium to high quality with a low risk of bias. However, comparing these studies is challenging due to heterogeneity in the definitions of cachexia, study designs, interventions, and outcomes. A significant issue is the varied definitions of cachexia. A definition based solely on BMI has become inadequate in light of the global obesity epidemic and the evolving understanding of metabolic changes preceding measurable body weight changes. Contemporary definitions of cachexia aim to identify signs and symptoms of malnutrition, clinical–laboratory evidence of inflammation, and loss of muscle mass and function [14,48]. Furthermore, while hypoalbuminemia (Alb < 35 g/L) is associated with malnutrition and advanced chronic disease, it can also result from other conditions not progressing to cachexia (e.g., nephrotic syndrome, protein-losing enteropathy) [49]. Suno et al. and Chiba et al. adopted the EPCRC criteria, classifying patients as cachectic with a stable body weight loss of more than 5% in the last six months, a BMI of less than 20 kg/m² with continuous weight loss of more than 2%, or sarcopenia with continuous weight loss of more than 2% [50]. Nevertheless, this definition does not consider inflammatory status, which is a key mechanism in cachexia syndrome. The diversity in definitions of cachexia, which does not adequately capture the complex metabolic and inflammatory processes underlying the syndrome, highlights the need for more comprehensive criteria that include signs of inflammation in addition to the traditional markers of malnutrition and muscle wasting.

Another limitation is the variation in study designs, making it difficult to compare results from retrospective and prospective studies with different sample sizes. For instance, the study by Nomura et al. included only 18 patients [25], whereas Barratt et al. enrolled 620 patients [19].

The heterogeneity in selected interventions also complicates comparisons. Six studies investigated opioid conversion to or from TDF, while four administered TDF patches without mentioning previous opioid use. Opioid rotation can result in variable clinical responses and serum drug concentrations due to individual pharmacodynamic and pharmacokinetic parameters. Polymorphisms in opioid receptors contribute to variable responses with receptor subtype densities and receptor–effector relationships changing after opioid exposure and/or disease progression [51]. Incomplete cross-tolerance between opioids, where tolerance to one opioid does not fully extend to another, also affects analgesic efficacy and side effects [52]. In studies where cachectic patients were not switched from other opioids, there is variability in the timing of clinical and laboratory evaluations post-TDF application. Suno et al. collected blood samples during the steady state of fentanyl plasma concentration after at least 3 days of 24 h TDF patch application [27], whereas Heiskanen et al. collected samples 3 days after applying the first 72 h TDF patch, before reaching steady state [21]. Pain typically increases on the third day of a 72 h TDF patch, often leading to dose adjustments or switching to a 48 h interval [53]. Lower BMI (<18.5) has been associated with more frequent 48 h patch applications due to end-of-dose failure [54]. A study comparing 72 h TDF with 24 h TDF found better pain control with the once-daily patch in patients whose 72 h TDF did not last the full duration [55]. Thus, a 24 h TDF regimen could be a reasonable choice in cachectic patients. None of the studies reviewed evaluated the response to TDF according to the intensity of chronic pain, the pathogenic mechanism, the use of adjuvant medications, or the breakthrough pain (BTP) response. The variability in intervention protocols, particularly with regard to opioid rotation and the timing of TDF administration, underscores the need for standardized approaches to evaluate fentanyl efficacy, tolerability, and PK in patients affected by cachexia.

Moreover, another limitation is the variability in selected outcomes. Most studies evaluated one or more primary outcomes, except Barratt et al. and Kuip et al., who assessed only secondary outcomes [19,29]. Among the primary outcomes, the pain scales NRS and VAS were the most commonly used. A review showed good correlation between VAS, Verbal Rating Scale (VRS) or Verbal Descriptor Scale (VDS), and NRS, although VAS is more difficult to evaluate, especially in elderly patients, those with cognitive impairments, communication difficulties, and minority groups [56]. One study defined criteria for successful or partially successful opioid rotation, which was not universally applicable since not all studies included TDF switching. The same article evaluated the effect of opioid rotation on ESAS and SDS [28]. ESAS assesses 10 major symptoms (rated from 0 to 10) common in cancer patients over the previous 24 h: pain, fatigue, nausea, depression, anxiety, drowsiness, shortness of breath, appetite, insomnia, and well-being [57]. SDS is the sum of all ESAS symptoms except insomnia [58]. Both tools include an NRS pain evaluation and provide an analysis of symptoms in advanced-stage disease, modifiable by effective antinociceptive therapy and potentially exacerbated by its toxicity, making them valid for assessing TDF efficacy and safety in cachectic patients. Moryl et al. collected data on pain relief, worst pain, and least pain, defined as patient-reported outcomes (PROs). According to the US Food and Drug Administration (FDA), PROs are direct patient reports on their health status without clinician interpretation, which are useful for monitoring therapy adverse effects, symptom control, and understanding patient perceptions of therapy impact on effectiveness [59]. PROs are currently underreported in advanced cancer cachexia trials but could be valuable for assessing TDF in cachectic patients [60]. Among the secondary outcomes, plasma fentanyl concentration and TDF dose were the most commonly used. TDF dose provides a quantitative, indirect assessment of TDF efficacy. Consistent with the findings of Heiskanen et al. and Chiba et al. [21,31], a study involving 1154 patients admitted to a PC unit found the median TDF dose on admission was three times higher than that of orally treated patients (median MED 180 mg of TDF vs. 60 mg of oral morphine), suggesting high doses of fentanyl on admission often had little benefit and significant side effects [61]. The variability in selected outcomes, particularly the use of different pain scales and the underreporting of PROs, highlights the importance of uniform assessment tools to ensure consistent evaluation of TDF in cachectic patients.

## 5. Conclusions

In conclusion, our scoping review reveals that regarding the primary objective, there is currently insufficient evidence in the literature to ascertain whether the efficacy and tolerability of TDF in cachectic patients differ from those in non-cachectic populations. As a result, it is not possible to provide definitive recommendations for prescribing or adjusting the dose of TDF in patients with cachexia. Concerning the secondary objective, the existing literature is inadequate to determine whether any specific clinical effect of TDF in cachectic patients can be attributed to pharmacokinetic differences.

Therefore, a prospective clinical trial with a substantial sample size of patients meeting a comprehensive and standardized definition of cachexia is warranted. This trial should aim to assess the impact of TDF on validated clinical outcomes such as pain and symptom rating scales as well as relevant pharmacokinetic parameters including serum concentrations of fentanyl and its metabolites. In addition, the study should investigate how cachexia-related pathophysiological changes affect drug absorption, distribution, metabolism and excretion by assessing markers of skin dryness, inflammatory status, hepatic and renal clearance.

## Figures and Tables

**Figure 1 cancers-16-03094-f001:**
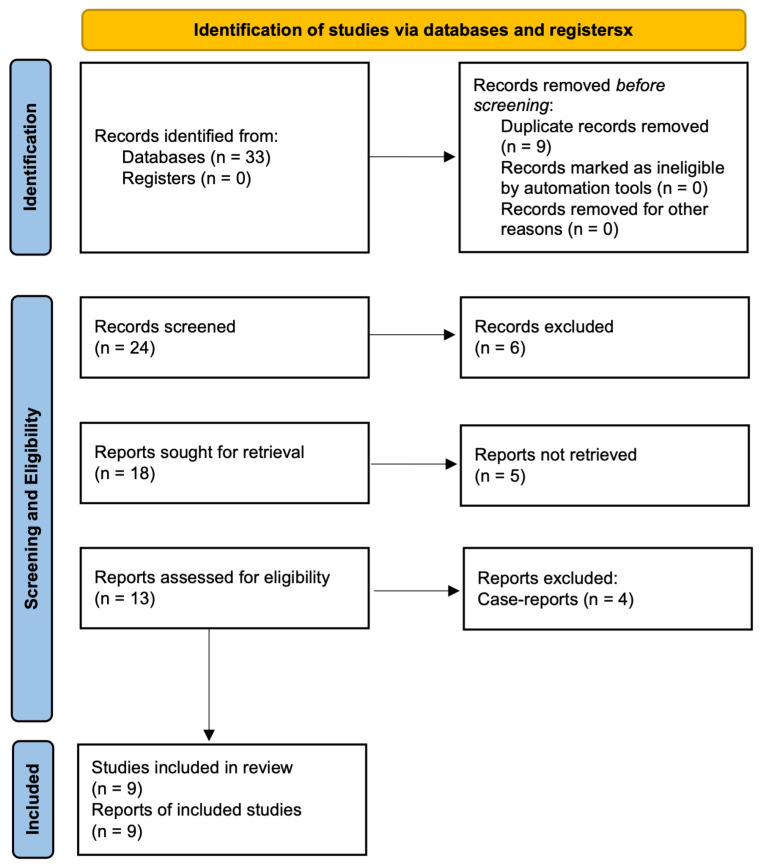
Preferred Reporting Items for Systematic Reviews and Meta-Analyses Extension for Scoping Reviews (PRISMA-ScR) flow diagram.

**Table 1 cancers-16-03094-t001:** Characteristics of the studies included in the review.

	Study Design	Participants	Intervention(s)	Outcome(s)
Heiskanen, 2009 [21]	Prospective observational study	Adult cancer patients divided by BMI (*n* = 20)	Other opioids → 72 h TDF (for 3 days at least)	[I] VAS [II] TDF dose, plasma fentanyl concentration
Nomura, 2013 [25]	Prospective observational study	Adult cancer patients divided by BMI and serum Alb (*n* = 18)	Fentanyl CII → TDF	[I] NRS, rescue events, opioid-induced side effects [II] Plasma fentanyl concentration
Barratt, 2013 [19]	Prospective observational study	Adult cancer patients exposed to different BMI and serum Alb levels (*n* = 620)	TDF for 3 days at least	[II] Plasma fentanyl and norfentanyl concentration, MR
Takahashi, 2014 [26]	Retrospective observational study	Adult cancer patients divided by NRS-2002 questionnaire (*n* = 92)	Oxycodone → TDF	[I] NRS
Suno, 2015 [27]	Prospective observational study	Adult patients classified according to EPCRC criteria (*n* = 21)	24 h TDF for 3 days at least	[I] VAS, opioid-induced side effects [II] Plasma fentanyl concentration
Reddy, 2016 [28]	Retrospective observational study	Adult cancer patients divided by BMI and serum Alb levels (*n* = 129)	Other opioids → TDF	[I] ESAS, SDS, successful/partial successful OR, opioid-induced side effects [II] MED
Kuip, 2018 [29]	Prospective observational study	Adult cancer patients divided by BMI (*n* = 88)	72h TDF for 8 days at least	[II] Plasma fentanyl concentration
Moryl, 2019 [30]	Prospective observational study	Adult cancer patients divided by BMI (*n* = 240)	72 h TDF	[I] Worst and least pain intensity scores, percent pain relief [II] TDF dose
Chiba, 2020 [31]	Retrospective observational study	Adult patients classified according to EPCRC criteria (*n* = 77)	24 h TDF for 3 days at least	[I] NRS [II] TDF dose, MED, TEWL

The outcomes are categorized in primary [I] and secondary [II]. Abbreviations: TDF, Transdermal Fentanyl; NRS, Numeric Rating Scale; MED, Morphine Equivalent Dose; BMI, body mass index; VAS, Visual Analogue Scale; Alb, Albumin; CII, Continuous Intravenous Infusion; MR, Metabolic Ratio; NRS-2002, Nutritional Risk Screening 2002; ESAS, Edmonton Symptom Assessment Scale; SDS, Symptom Distress Score; OR, opioid rotation; EPCRC, European Palliative Care Research Collaborative; TEWL, Transepidermal Water Loss.

**Table 2 cancers-16-03094-t002:** Impact of cachexia on primary and secondary outcomes: comparison between cachectic patients used as reference and non-cachectic patients.

	Primary Objective(s)	Secondary Objective(s)	Hypothesized Pathophysiological Mechanism
Heiskanen, 2009 [21]	= VAS	↑ TDF dose ↓ Plasma fentanyl concentration	Changes in skin permeability (xerosis) resulting in a decreased TDF absorption rate.
Nomura, 2013 [25]	= NRS = Rescue events = Opioid-induced side effects	= (BMI) / ↓ (Alb) Plasma fentanyl concentration	Hypoalbuminemia may cause an undetectable edema and reduced skin permeability resulting in a decreased TDF absorption rate.
Barratt, 2013 [19]	-	↓ Plasma fentanyl concentration ↑ Plasma norfentanyl concentration ↑ MR	No significant differences were observed between cachectic and non-cachectic patients.
Takahashi, 2014 [26]	↑ NRS	-	Changes in skin permeability (cutaneous dryness) resulting in a decreased TDF absorption rate.
Suno, 2015 [27]	= VAS Opioid-induced side effects not reported	↑ Plasma fentanyl concentration	Reduction in eGFR and cachexia-related inflammation which downregulates CYP3A4 may increase fentanyl plasma concentrations.
Reddy, 2016 [28]	= ESAS and SDS = Successful/partial successful OR = Opioid-induced side effects	= MED	No significant differences were observed between cachectic and non-cachectic patients.
Kuip, 2018 [29]	-	= Plasma fentanyl concentration	No significant differences were observed between cachectic and non-cachectic patients.
Moryl, 2019 [30]	↑ Percent pain relief ↓ Least pain	↓ TDF dose	No pathophysiological mechanism was hypothesized.
Chiba, 2020 [31] *	↓ NRS	↑ TDF dose ↑ MED ↓ TEWL	IL-4 elevation from cancer cachexia may decrease ceramide in the skin surface impairing stratum corneum water-holding capability (cutaneous dryness) and reducing the TDF absorption rate.

↑ Increased values of the objective(s) in cachectic patients compared to non-cachectic patients. ↓ Decrease values of the objective(s) in cachectic patients compared to non-cachectic patients. = No difference between cachectic and non-cachectic patients in the considered objective(s). * Considering the conversion from oral oxycodone/morphine to TDF. Abbreviations: VAS, Visual Analogue Scale; TDF, Transdermal Fentanyl; NRS, Numeric Rating Scale; BMI, body mass index; Alb, Albumin; MR, Metabolic Ratio; eGFR, estimated Glomerular Filtration Ratio; ESAS, Edmonton Symptom Assessment Scale; SDS, Symptom Distress Score; OR, opioid rotation; MED, Morphine Equivalent Dose; TEWL, Transepidermal Water Loss; IL-4, interleukin-4.

**Table 3 cancers-16-03094-t003:** Risk of bias analysis using NIH quality assessment tool for observational studies.

	Q1	Q2	Q3	Q4	Q5	Q6	Q7	Q8	Q9	Q10	Q11	Q12	Q13	Q14	Grading
Heiskanen, 2009 [21]															Low risk
Nomura, 2013 [25]							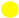								Low risk
Barratt, 2013 [19]							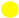								Low risk
Takahashi, 2014 [26]												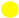			Fair risk
Suno, 2015 [27]															Fair risk
Reddy, 2016 [28]															Low risk
Kuip, 2018 [29]															Low risk
Moryl, 2019 [30]												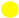			Low risk
Chiba, 2020 [31]															Low risk

Green circle represents low bias, yellow circle represents unclear and red indicates high bias. Abbreviations: NIH, National Institute of Health.

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
