# Peer review of "Transdermal Fentanyl in Patients with Cachexia—A Scoping Review"

_cancers, 2024, doi:10.3390/cancers16173094_

Round 1

Reviewer 1 Report

Comments and Suggestions for Authors

This scoping review analyzes the impact of cachexia on TDF pharmacokinetics (PK). The analgesic effect and tolerability of TDF in patients with viral disease were evaluated, and cachexia features that may affect fentanyl PK were identified. The paper is rich in workload and has certain research significance. However, there are still some problems that need to be modified.

1.     Lines 28-30 can be divided into two sentences describing the characteristics and transdermal absorption process of TDF.

2.     CYP3A4 and CYP3A5 in line 33 should be given full or brief explanations.

3.     There are some problems with the format of Table 2, please modify it.

4.     The unit of concentration of TDF in line 285 is g/h, and it is suggested to consider using μg/h or mg/h.

5.     The discussion part of this paper is rich in content, but it is less segmented. It is suggested to reschedule according to the content.

6.     It is recommended to add a concluding statement at the end of each paragraph in the discussion section of the article.

7.     The conclusion of the paper does not seem to involve the objective stated in the abstract, so it is suggested to add related descriptions.

Comments on the Quality of English Language

The content, structure and expression of the article are clear, but there are still several aspects that can be improved and improved.

1.     “Fentanyl is a synthetic µ-opioid receptor agonist with an affinity 80-100 times greater 18 than morphine” should be changed “Fentanyl is a synthetic µ-opioid receptor agonist with an affinity of 80-100 times greater 18 than morphine”.

2.     “Some studies have suggested that transdermal fentanyl (TDF) is associated with a lower incidence of adverse effects such as constipation, nausea, vomiting, and daytime drowsiness” should be changed “Some studies have suggested that transdermal fentanyl (TDF) is associated with a lower incidence of adverse effects such as constipation, nausea, vomiting and daytime drowsiness”.

3.  It is suggested to add transition words and linking words between sentences and paragraphs to make the essay more coherent.

Author Response

Comment 1:  Lines 28-30 can be divided into two sentences describing the characteristics and transdermal absorption process of TDF.

Response 1: We have divided those lines (L39-41) into two sentences. 

Comment 2: CYP3A4 and CYP3A5 in line 33 should be given full or brief explanations.

Response 2 (L48): We provided a brief explanation of CYP3A4 and CYP3A5 by identifying them as part of the cytochrome P450 family.

Comment 3: There are some problems with the format of Table 2, please modify it.

Response 3: We were not able to correctly modify Table 2 using LaTeX code. We will upload a Word file containing all the tables from the article.

Comment 4: The unit of concentration of TDF in line 285 is g/h, and it is suggested to consider using μg/h or mg/h.

Response 4 (L298): We have corrected the unit of concentration of TDF in line 285 to μg/h.

Comment 5: The discussion part of this paper is rich in content, but it is less segmented. It is suggested to reschedule according to the content.

Response 5: We have reorganized the discussion section of the article by separating it into 7 paragraphs.

Comment 6: It is recommended to add a concluding statement at the end of each paragraph in the discussion section of the article.

Response 6: We have included a concluding statement sentence for each paragraph, as it follows:

  • L306-309: These cases highlight the urgency to better comprehend how cachexia interacts with TDF exposure-response relationship and to ensure effective pain management while minimizing severe adverse effects, particularly in the context of frail and often end-of-life patients.
  • L327-329: Given these conflicting findings, it is currently impossible to provide a definitive recommendation on whether, when, or how to prescribe TDF in these patients. 
  • L371-374: Given the limited evidence, the potential impact of cachexia on TDF PK appears to be multifaceted, involving altered skin permeability and systemic inflammatory responses that could modulate both drug absorption and metabolism, leading to significant variability in circulating fentanyl levels.
  • L390-393: The diversity in definitions of cachexia, which does not adequately capture the complex metabolic and inflammatory processes underlying the syndrome, highlights the need for more comprehensive criteria that include signs of inflammation in addition to the traditional markers of malnutrition and muscle wasting.
  • L394-397: the paragraph is really short. We thought that a concluding statement sentence is not necessary. 
  • L418-421: The variability in intervention protocols, particularly with regard to opioid rotation and timing of TDF administration, underscores the need for standardized approaches to evaluate fentanyl efficacy, tolerability, and PK in patients affected by cachexia. 
  • L450-452: The variability in selected outcomes, particularly the use of different pain scales and the underreporting of PROs, highlights the importance of uniform assessment tools to ensure consistent evaluation of TDF in cachectic patients.

Comment 7: The conclusion of the paper does not seem to involve the objective stated in the abstract, so it is suggested to add related descriptions.

Response 7: We have re-writed primary and secondary objectives in the introduction to promote congruence with the conclusion, as follows (L71-76): This scoping review aims to analyze and categorize the evidence concerning TDF administration in patients with cachexia, irrespective of the underlying pathology. The primary objective is to assess whether the efficacy and tolerability of TDF in cachectic patients differ from those observed in non-cachectic patients. The secondary objective is to determine whether the specific clinical effects of TDF in cachectic patients can be attributed to pharmacokinetic differences. 

Then, we modified the conclusion in order to involve the objectives stated, as follows (L454-460): In conclusion, our scoping review reveals that, regarding the primary objective, there is currently insufficient evidence in the literature to ascertain whether the efficacy and tolerability of TDF in cachectic patients differ from those in non-cachectic populations. As a result, it is not possible to provide definitive recommendations for prescribing or adjusting the dose of TDF in patients with cachexia. Concerning the secondary objective, the existing literature is inadequate to determine whether any specific clinical effect of TDF in cachectic patients can be attributed to pharmacokinetic differences.

Furthermore, we made some modifications in the abstract. 

Quality of English Language: We tried to improve English Language.

Reviewer 2 Report

Comments and Suggestions for Authors

 This manuscript attempted to review the existing literature about the impact of cachexia on the efficacy, tolerability and pharmacokinetics of transdermal fentanyl. The authors failed to make definitive conclusions on this subject from the current literature and suggested a prospective clinical trial to answer this question. As a whole, the review is extensive and informative and I have no comments except to recommend its acceptance for publication with no changes.

Author Response

Thank you for your recommendation.

Reviewer 3 Report

Comments and Suggestions for Authors

The Authors perform a scoping review of the literature on the topic of fentanyl in patients with cachexia in an attempt to draw a conclusion of therapeutic efficacy.  The Authors conduct the literature review well and the analysis is also well done, using the NIH Quality Assessment tool.   Unfortunately, the review of the selected studies was insufficient to provide any definitive recommendations on the use of TDF in patients with cachexia.  1) The Authors do identify the variability in the methods used to diagnose cachexia, which potentially leads to the variability.

2) The Authors also use this inconclusive analysis as a way to suggest  a prospective trial with patients using a standardized method of diagnosing cachexia as a way to determine the therapeutic efficacy of TDF in this patient population.  This is where I think the Authors can strengthen the manuscript.  Can the Authors enhance this section and provide more specific details or considerations for a trial design?  Such things that could be further considered include: cancer type; cancer stage; sex; age.  Since the analysis in its current form is inconclusive,  providing expanded suggestions for a trial to determine this effect would strengthen the paper. 

Author Response

Comment 1: can the Authors enhance this section and provide more specific details or considerations for a trial design?

Response 1: We appreciate the reviewer's suggestion and agree that offering more detailed recommendations for a trial to assess this effect would enhance the paper. However, given the complexity and variability of potential study designs, it is challenging to propose a detailed hypothetical trial. A possible approach could involve a parallel-cohort design, with one cohort comprising cachectic patients and the other comprising non-cachectic patients. This design might be well-suited to evaluate the differences in outcomes between these two populations.

Reviewer 4 Report

Comments and Suggestions for Authors

I have read and reviewed this manuscript with great interest and overall, from this reviewer's perspective, it is a study that has been well-planned and executed. Overall it is a study with refreshingly simple wording that is easy to understand. Other strengths of the manuscript that I can highlight are the following: the introduction provides sufficient background and includes pertinent references, the research design is adequate, and the methods are repeatable and correctly described, although the conclusions must be rewritten to be consistent with the stated aims.

Nevertheless, some points must be addressed to achieve publication quality. I have left some comments hoping that they can help the authors.

General comments

L32: Please provide data on bioavailability, Tmax, Cmax, AUC, and Vd of TDF.

L84: participants

L155: what medications were used to control adverse effects? Please clarify.

L156: what NSAIDs were used?

L157: for what purpose were CYP3A inhibitors-inducers used?

Table 1: write in brackets the reference number to which each citation used corresponds. In this same table, I suggest the authors describe in greater detail the outcomes of each cited article.

L378: I suggest the authors consider other factors in their discussion, for example, chronic pain. How does this factor influence the analgesic efficacy of TDF? And, from the literature reviewed regarding rescue analgesia, which analgesics or adjuvants have provided the best results?

L418: I also suggest the authors include in their discussion the effects that pure opioids such as fentanyl have on the immune response, emphasizing the effects that can be generated in the cancer patient. Likewise, it is necessary to include in your discussion the limitations and perspectives of your study.

L420: The ideas described in L420-431 could be included in the limitations and perspectives of your study, therefore, the conclusions should be rewritten to show congruence with what is described in the aims of the study.

Comments on the Quality of English Language

Minor editing of English language required.

Author Response

Comment 1 (L32): Please provide data on bioavailability, Tmax, Cmax, AUC, and Vd of TDF.

Response 1: We have included all the requested data in the introduction of the article.

Comment 2 (L84): participants

Response 2: Corrected (L100)

Comment 3 (L155): what medications were used to control adverse effects?

Response 3: We included this sentence (L167-169): Heiskanen et al. reported the use of laxatives for constipation and haloperidol for nausea, whereas Nomura et al. noted the occasional use of stool softeners, laxatives, or enemas.

Comment 4 (L156): what NSAIDs were used?

Response 4: Takahashi (2014) and Chiba (2020) did not report the NSAIDs used.

Comment 5 (L157): for what purpose were CYP3A inhibitors-inducers used?

Response 5: The study by Barratt (2013) aimed to analyze the relative contribution of various factors to the pharmacokinetics of transdermal fentanyl, including those that may interfere with hepatic metabolism, like CYP3A4 inhibitors-inducers. Inhibitors were categorized as strong, moderate, or weak, corresponding to a decrease in clearance of over 80%, 50–80%, and 20–50%, respectively. Inducers were differentiated between glucocorticoids and other inducing drugs. However, the specific CYP3A4 inhibitors and inducers used in the study were not reported.

Comment 6 (Table 1): Write in brackets the reference number to which each citation used corresponds. In this same table, I suggest the authors describe in greater detail the outcomes of each cited article.

Response 6: We placed the reference numbers in brackets in table 1 and 2. To provide a more detailed description of the outcomes in table 1, we divided the outcomes into primary and secondary categories.

Comment 7 (L378): I suggest the authors consider other factors in their discussion, for example, chronic pain. How does this factor influence the analgesic efficacy of TDF? And, from the literature reviewed regarding rescue analgesia, which analgesics or adjuvants have provided the best results?

Response 7: We included this sentence in the discussion (L414): None of the studies reviewed evaluated the response to TDF according to the intensity of chronic pain, the pathogenic mechanism, the use of adjuvant medications, or the breakthrough pain (BTP) response.

Comment 8 (L418): I also suggest the authors include in their discussion the effects that pure opioids such as fentanyl have on the immune response, emphasizing the effects that can be generated in the cancer patient. Likewise, it is necessary to include in your discussion the limitations and perspectives of your study.

Response 8: We included this sentence in the discussion (L369): Unfortunately, the studies included in our review did not examine the relationship between immune status, cachexia, and opioid use. Notably, an emerging body of evidence suggests that opioids may interact with immune system activity in cancer patients, particularly in those undergoing treatment with immune checkpoint inhibitors (ICIs). These aspects, which could be crucial for a comprehensive risk-benefit analysis of opioid therapies, should be considered in future research.

Additionally, we have included the appropriate bibliographic references.

Comment 9 (L420): The ideas described in L420-431 could be included in the limitations and perspectives of your study, therefore, the conclusions should be rewritten to show congruence with what is described in the aims of the study.

Response 9: We revised the conclusion to better align with the stated objectives, as follows (L454-460): In conclusion, our scoping review reveals that, regarding the primary objective, there is currently insufficient evidence in the literature to ascertain whether the efficacy and tolerability of TDF in cachectic patients differ from those in non-cachectic populations. As a result, it is not possible to provide definitive recommendations for prescribing or adjusting the dose of TDF in patients with cachexia. Concerning the secondary objective, the existing literature is inadequate to determine whether any specific clinical effect of TDF in cachectic patients can be attributed to pharmacokinetic differences.

Moreover, we appreciate the reviewer's suggestion; however, we believe it may not be fully applicable in this case, as the information referenced in these lines was obtained retrospectively as part of our study results, rather than indicating a limitation of the study.

Round 2

Reviewer 1 Report

Comments and Suggestions for Authors

All issues were addressed.

Reviewer 3 Report

Comments and Suggestions for Authors

The Authors have revised the paper appropriately.